


# Enhanced toxicity of aerosol in fog conditions in the Po Valley, Italy

Stefano Decesari[1], Mohammad H. Sowlat[2], Sina Hasheminassab[2], Silvia Sandrini[1], Stefania Gilardoni[1], Maria Cristina Facchini[1], Sandro Fuzzi[1], Constantinos Sioutas[2]

[1]Institute of Atmospheric Sciences and Climate of the National Research Council of Italy, Bologna, Italy
[2]Department of Civil and Environmental Engineering, University of Southern California, Los Angeles, CA, USA

*Correspondence to*: Stefano Decesari (s.decesari@isac.cnr.it)

**Abstract**. While numerous studies have demonstrated the association between outdoor exposure to atmospheric particulate matter (PM) and adverse health effects, the actual chemical species responsible for PM toxicological properties remain a subject of investigation. We provide here reactive oxygen species (ROS)

activity data for PM samples collected at a rural site in the Po Valley, Italy, during the fog season (i.e., November-March). We show that the intrinsic ROS activity of Po Valley PM, which is mainly composed of biomass burning and secondary aerosols, is comparable to that of traffic-related particles in urban areas. The airborne concentration of PM components responsible for the ROS activity decreases in fog conditions, when water-soluble species are scavenged within the droplets. Thanks to this partitioning effect of fog, the

measured ROS activity of fog water was contributed mainly by water-soluble organic carbon (WSOC) and secondary inorganic ions rather than by transition metals. We found that the intrinsic ROS activity of fog droplets is even greater (> 2.5 times) than that of the PM on which droplets are formed, indicating that redox-active compounds are not only scavenged from the particulate phase, but are also produced within the droplets. Therefore, even if fog formation exerts a scavenging effect on PM mass and redox-active

compounds, the aqueous-phase formation of reactive secondary organic compounds can eventually enhance ROS activity of PM when fog evaporates. These findings, based on a case study during a field campaign in November 2015, indicate that a significant portion of airborne toxicity in the Po Valley is largely produced by environmental conditions (fog formation and fog processing) and not simply by the emission and transport of pollutants.

**1 Introduction**

There is a rapidly growing body of epidemiological evidence identifying major health impacts associated with population exposure to airborne particulate matter (PM), including, but not limited to, respiratory and cardiovascular diseases, as well as neurodegenerative effects (Pope et al., 2002; Pope et al., 2004; Dockery and Stone, 2007; Davis et al., 2013;Gauderman et al., 2015). Even if the international air quality standards

for atmospheric PM are based on mass concentrations (PM10 and PM2.5 for particles of diameter below 10 or 2.5 μm, respectively), the WHO acknowledges that it is likely that not every PM component is equally important in causing these health effects (WHO, 2007, 2013). Recently, research on traffic-related PM has provided a first epidemiological evidence of the links between adverse health effects and PM chemical composition (Janssen et al., 2011), in line with the results of several toxicological studies (Nel, 2005; Cassee



et al., 2013; Bates et al., 2015; Fang et al., 2015), while outside urban areas, the characterization of PM and associated health effects is sparse. During transport in the atmosphere, particles emitted by traffic are progressively diluted and eventually transformed by chemical processes that enrich them in secondary components (Zhang et al., 2007; Crippa et al., 2014). Moreover, the oxidation of reactive volatile organic

compounds (VOCs) supplies a range of water-soluble compounds (e.g., formaldehyde, glyoxal), partitioning into the particulate phase at high relative humidities (deliquescent aerosols and cloud/fog droplets), where they can be further oxidized into new secondary organic compounds (so called "aqueous SOA") (Ervens et al., 2011). The current understanding of the health impacts of PM secondary components is certainly much more limited than for traffic-related aerosols. Although recent chamber studies have indicated increased

oxidative potential of PM emitted from a variety of sources (e.g., combustion or biomass burning) after undergoing oxidation and/or secondary formation (Antinolo et al., 2015; McWhinney et al., 2011, 2013), there is a lack of direct field toxicological observations, which is partly caused by the difficulty of disentangling the secondary PM fraction from other oxidized organic material of primary nature in field conditions. In this study, we investigate the toxicological properties of PM at a polluted rural site, where the presence of fog is

responsible for aerosol scavenging and deposition but also for secondary aerosol formation through aqueous-phase processes.

The measurements were carried out in the Po Valley, Italy, where radiation fogs frequently occur during the cold season (up to 25% of the time in fall-winter months in rural areas, according to recent studies (Giulianelli et al., 2014). With approximately 20 million inhabitants (~30% of the Italian population), the Po Valley area

has the highest population density across the country, and ranks among the top European "pollution hotspots" in terms of mortality attributable to PM exposure (Kiesewetter and Amann, 2014). We provide here the first measurements of the redox activity (as a proxy for toxicological potential) of fog water and interstitial (i.e., unscavenged) aerosols. The toxicological assay used in this study is capable of quantifying the oxidative potential associated with ambient PM, initiated via generation of reactive oxygen species (ROS) due

to the interaction of target cells with redox-active components of ambient PM (Landreman et al., 2008; Daher et al., 2012; Verma et al., 2012). We show that ROS, in the rural Po Valley PM, occur in concentrations comparable to that of a PM in a megacity, and that the ROS levels are even amplified in fog water with respect to the PM fraction scavenged within fog droplets. Results from the present study provide a basis for prospective epidemiological programs to evaluate how fog scavenging/processing of PM impacts on human

health.

## 2 Methods

### 2.1 Measurement site

Fog samples were collected from 30 November to 30 December 2015  at the meteorological station Giorgio Fea in San Pietro Capofiume (44°39'15" latitude, 11°37'29" longitude), a rural site located 30 km northeast

of Bologna (Italy) in the eastern part of the Po Valley (northern Italy). From 30 November to 4 December, an




intensive observation period was scheduled, with the concurrent sampling of fog and aerosol samples and the deployment of a HR-ToF-AMS (Aerodyne Research) for online aerosol measurements. During the sampling campaign, a total of 6 aerosol samples and 16 fog samples were collected. Additionally, fog samples collected after the intensive observation period (i.e., after 4 December) were pooled in groups of two or

three for the analysis of metals and oxidative potential.

**2.2 Aerosol and fog-water sampling**

Aerosol samples were collected by a PM1 Tecora Echo High Volume sampler equipped with a PM1 Digitel sampling head, operating at 500 L min$^{-1}$ flow. One daytime and one nighttime sample were collected every day from 9:00 to 18:00 LT and from 18:00 to 9:00 LT respectively, on prewashed and prebaked Pall quartz

fiber filters. Samples were wrapped with aluminum foil, zipped in plastic bags, and stored in freezer at −20°C until analysis. Fog samples were collected using an automated, computer driven active string collector described in Fuzzi et al. (1997). A Particulate Volume Monitor PVM-10 (Gerber, 1991), used to determine fog liquid water content (LWC) at 1 min time resolution, was used to activate the string collector. A LWC threshold of 0.08 g m$^{-3}$ was chosen for the activation as an indicator of fog presence, which roughly

corresponds to a 200 m visibility. Concentrations of analytes in fog samples, expressed in μg mL-1, were converted to μg m$^{-3}$ by multiplying by the LWC of each sample.

**2.3 Chemical analysis of water-soluble components**

Prior to chemical analysis a quarter of each aerosol quartz-fibre filter was extracted with 10 mL of 18-MΩ Milli-Q water by sonication for 30 minutes. Liquid extracts were filtered to remove quartz residues, then

analyzed for Water Soluble Organic Carbon (WSOC) with a total organic carbon analyzer (Shimadzu TOC-5000A). Fog samples were filtered on 47 mm quartz-fibre filters (Whatman, QMA grade) within few hours after collection to remove suspended particulate. Conductivity and pH measurements were carried out immediately, then samples were stored frozen until further analysis. WSOC was determined in the same way as for aerosol filter extracts, in addition the ionic composition of fog was determined by two Thermofisher

ICS_2000 ion chromatographers equipped with an IonPac AS11 2x250 mm separation column for anions and IonPac CS16 3x250 mm Dionex separation column, self-regenerating suppressors and KOH and MSA as eluents for anions and cations respectively. In both cases a gradient elution allowed the separation and detection of both inorganic and organic anions and cations.

**2.4 Total elemental analysis of aerosol and fog samples**

The magnetic sector inductively coupled plasma mass spectrometry (SF-ICP-MS; Thermo-Ginnigan Element 2) unit was used for the total elemental analysis of the aerosol and fog samples. Detailed information about the procedure can be found elsewhere (Zhang et al., 2008; Okuda et al., 2014). Briefly, the samples are first digested in an automated microwave-aided (oven-aided in case of fog samples) digestion system (Milestone ETHOS+) using a mixture of ultra-high-purity acids (1.0 mL of 16 M nitric acid and 0.25 mL of 12 M



hydrochloric acid). Then, the digestates are analyzed by SF-ICPMS after being diluted to 15 mL with high-purity water (18 mΩ). A combination of the SF-ICPMS instrumental analysis, method blanks, and digestion recoveries were used to estimate the analytical uncertainties. A total of 49 elements were determined using this method. Further details of the analytical method can be found in Zhang et al. (2008) and Okuda et al.,

5    (2014).

**2.5 ROS analysis**

The ROS assay, an in vitro exposure assay of PM extracts to rat alveolar macrophage cells (cell line NR8383)(Landreman et al., 2008), was used as a measure of PM toxicity of the samples. In this method, samples (fog and filters) were first extracted using an initial sonication period for 15 min with high-purity

Milli-Q (18 mΩ) water. The samples are then continuously agitated in the dark at room temperature for 16 h, followed by another 15-min sonication and 1-min agitation by a vortex mixer. A portion of the sample suspension was passed through 0.22 μm polypropylene syringe filters (referred to as filtered samples, representing WS components), while the other portion of the sample suspension underwent the ROS assay without filtration (referred to as the unfiltered samples)(Shafer et al., 2016).

In the in vitro exposure and ROS detection step, the membrane-permeable 2',7'-dichlorodihydro- fluorescein diacetate (DCFH-DA) probe gets deacetylated by cytoplasmic esterases to 2',7'-dichlorodihydro- fluorescein (DCFH) upon entering the macrophage cell. In the presence of oxidizing species (for example, the ROS species generated due to exposure to toxic PM components), DCFH is converted to its fluorescent form, which is DCF. In the next step, an M5e microplate reader (Molecular Devices, CA, USA) is used to determine the

fluorescence intensity of each cell after the exposure at 488 nm excitation and 530 nm emission. The raw fluorescence data were corrected using blanks and normalized using zymosan (ZYM) positive controls (i.e., β-1,3 polysaccharide of D-glucose from Sigma Aldrich, MO, USA) (Shafer et al., 2016). Filter blanks were used to account for the potential impact of the biologically active quartz filters on the ROS results. Finally, the overall per-mass oxidative potential (representing the inherent toxicity) of samples was reported in units of

Zymosan equivalents. The per-volume ROS activity (i.e., normalized by the volume of air samples) of samples was also calculated via multiplying the mass-based ROS by the PM mass concentration of a given sample. This provides a measure of the actual toxicity of the air inhaled by individuals, which is a very important parameter in exposure assessment studies (Wittkopp et al., 2013; Zhang et al., 2007; Delfino et al., 2010).

**2.6 Statistical analyses**

Spearman rank correlation was used to explore the association of individual components of the aerosol and fog with the ROS activity. We also applied the principal component analysis (PCA) to the ambient concentrations of the chemical components of the aerosols in order to identify the source factors that contribute to PM levels and ROS activity. In this analysis, the VARIMAX-normalized rotation approach was employed to identify the uncorrelated source factors (Henry, 1987). Additionally, we also applied the multi-

linear regression (MLR) analysis to identify the source factors (as represented by the relevant species) that





mostly contribute to ROS activity of aerosols and fog water. In this analysis, several combinations of species with high loadings in the PCA were regressed against the ROS levels, and the ones leading to the highest R2 value were kept in the final solution. The statistical significance was evaluated at both P values of <0.1 and <0.05.

**3 Results**

**3.1 ROS activity of fog water and aerosol samples**

When fog forms, aerosol particles are selectively scavenged into fog droplets, with the smaller and more hydrophobic particles left unscavenged as interstitial aerosol. This process tends to be selective with respect to specific chemical components ("partitioning") and represents a useful way to study the toxicological

properties of externally-mixed PM components under field conditions. Past studies in the Po Valley (Facchini et al., 1999; Gilardoni et al., 2014) highlighted that fog scavenging efficiency for different chemical components of atmospheric particles is related to their water solubility, with higher scavenging efficiencies for water-soluble (WS) species (e.g., inorganic ions and water-soluble organic carbon, WSOC) and lower scavenging efficiencies for hydrophobic compounds (e.g., elemental carbon and several metals) (Gilardoni et

al., 2014). Therefore, during fog episodes, fog droplets are enriched in WS components, while water-insoluble (WI) species dominantly partition into interstitial aerosols (Hallberg et al., 1992; Fuzzi et al., 1988). Results from the present study are in good agreement with the published literature on the effect of fog scavenging on the aerosol properties: Figures S2 and S3 in Supporting Information indicate that organic as well as inorganic WS components (e.g. MSA, oxalate, nitrate, and sulfate) and metals with high WS fractions

(including Ca, Na, and Mg) have the highest scavenging rate, whereas other combustion-related species with much lower solubility (i.e. Fe, Cr, and Zn) are scavenged much less efficiently. Metals can be also scavenged by fog but not to the same extent as secondary organic and inorganic species, resulting in higher metal enrichment of the interstitial aerosol. The results on partitioning of WSOC between interstitial aerosols and fog water are presented in Figure 1c,d. Contrary to the chemical species discussed above, WSOC is a complex

mixture of chemical species originating from a variety of sources. According to previous studies in the region (Gilardoni et al., 2014; 2016), oxidized particulate organic compounds include biomass burning products as well as secondary organic species, which also account for the products of transformation of biomass burning compounds upon fog processing. As can be seen in Figure 1c, the mass fraction of WSOC in PM1 was comparable between daytime and interstitial aerosols, analogously to what observed for some of the WS

ionic species, like sulfate and oxalate, and contrary to others (e.g., nitrate and ammonia) which are considerably depleted in interstitial particles with respect to daytime aerosol. However, the airborne concentration of WSOC in daytime aerosol was approximately two times higher than that of interstitial aerosols (Figure 1c). Additionally, the concentration of WSOC was much higher in the fog water than in daytime and nighttime aerosols, both on a per-mass and per-volume basis.



We present here for the first time results on fog scavenging effects on ROS activity of the aerosol. The average ROS activity of daytime (i.e, without fog) and nighttime (in-fog as well as interstitial) aerosol bulk extracts and for fog water is reported in Figure 1a,b. On a per m$^3$ of air volume basis, the ROS activity of fog water and daytime aerosol are quite comparable and about four times greater than that of interstitial aerosol.

However, per PM mass ROS activities of the daytime and interstitial aerosols are very similar (1877±849 and 1870±1229 microgram Zymosan/mg of PM, respectively), while the ROS activity of fog water was considerably higher (5194±610 MicrogramZymosan/mg of PM) than that of both types of aerosols. The variability of ROS activity of PM in the Po Valley in the fog season is therefore complex. The extrinsic (on a per-volume basis) ROS activity in PM decreases upon fog scavenging similarly to total PM1 mass

concentrations (-70% and -66%, respectively) causing the toxicity of interstitial particles being comparable to that of the original aerosol population. In contrast, the ROS activity of fog water is clearly higher than the scavenged fraction of ROS activity of PM (calculated as the difference between daytime and interstitial PM activity). This trend reflects very closely that of WSOC levels (Figure 1c,d). The excess of WSOC mass in fog water with respect to daytime aerosol indicates that organic solutes in fog droplets do not originate simply

from aerosol scavenging, and that additional sources from aqueous-phase processing of adsorbed volatile organic compounds must be taken into account. Secondary organic aerosol (SOA) sources in fog water could therefore explain the "amplification" of ROS activity with respect to the parent aerosol population.

To further explore the difference in the toxicity of daytime and interstitial aerosols, the ROS activity of daytime and nighttime aerosols was also evaluated based on the filtered ROS analysis protocol, representing

the toxicity of the WS components (Figure 2a,b). As already mentioned, volume-based ROS activity of daytime aerosols was 3-5 times higher than that of interstitial aerosols, but this difference becomes even greater in the filtered extracts (a factor of 7, see Figure 2a). It should be noted that during the daytime, the per PM-mass filtered ROS activity of aerosols (which relates to the redox activity of only the WS fraction) is almost half of the unfiltered ROS levels (representing the redox activity of both WS and WI fractions).

However, during nighttime, the per PM mass ROS activity of the WS fraction contributes approximately only 25% to the overall per-mass ROS aerosol activity (Figure 2b). These results imply that during daytime, WS and WI PM fractions contribute roughly equally to the ROS activity of the aerosol, whereas during nighttime, the WI components (comprising mostly elements, metals and insoluble carbonaceous material) contribute to as much as 75% of the aerosols ROS activity. Therefore, even if daytime and interstitial aerosols exhibit the

same intrinsic toxicity (Figure 1b), this is driven by different chemical composition in the two aerosol populations. The most critical difference between daytime and interstitial (i.e., unscavenged) aerosols is that the latter are depleted of WS components, including WSOC (Figure 1) and inorganic aerosols (Figures S2 and S3). Therefore, as discussed above, the higher ROS activity of daytime aerosols compared to that of nighttime/interstitial aerosols must be attributed to WS components, including secondary ionic species and

WSOC. At nighttime, the aerosol ROS activity is mainly driven by WI components, as fog scavenging denudes particles of WS components (Figure S2), leaving the interstitial aerosols with higher mass fraction of WI





components. It is also noteworthy that the highest ROS activity was observed in the fog water, which is enriched in WS components compared to the daytime and interstitial aerosols.

**3.2 Association of ROS activity to PM and fog chemical components**

To investigate the main species driving the ROS activity in fog water, we performed Spearman rank correlation as well as Multilinear regression (MLR) analysis between per-volume concentrations of all species and per-volume ROS activity of fog water, the results of which are presented in Tables 1, 2, and in Table 3. As can be seen in Table 1, most of the WS species were strongly correlated with the ROS activity of fog. However, this does not necessarily mean that all of these species, including inorganic ions, are responsible for the aerosol redox activity. Previous studies have indicated that some of these associations with toxicologically innocuous species (e.g., inorganic ions) are observed because of the co-linearity of inorganic ions with important redox active species, including WSOC, and not because of the toxicity of these components per se (Ntziachristos et al., 2007; Cho et al., 2005; Verma et al., 2012). For instance, (Verma et al., 2012a) found strong positive correlations between the redox activity of quasi-ultrafine PM (PM0.18) and $NO_3^-$, $SO_4^{2-}$, $NH_4^+$, and WSOC; however, the authors concluded that the strong correlations observed between these inorganic ions and redox activity is because of their co-linearity with WSOC (as also shown in Table S1) rather than their own toxicity, as these components are not mechanistically active in these assays (26). This was also confirmed by the results of the MLR analysis (Table 3), demonstrating that WSOC alone explains 98% of the variability in the fog water ROS levels. We observed much smaller correlation coefficients between the ROS activity of fog water samples and elemental/metallic components, which have been shown to correlate with particle toxicity in earlier studies (Ntziachristos et al., 2007; Cho et al., 2005; Hu et al., 2010; Verma et al., 2012). This is due to the lower concentration of metals/elements in the fog water compared to daytime and interstitial aerosols, as shown in Figure S3.

To investigate the major PM chemical species that contribute to the ROS activity of daytime and interstitial aerosols, we performed a principal component analysis (PCA) followed by a linear correlation and a multilinear regression (MLR) analysis on the data pertaining to aerosols samples. It should be noted that, due to the limited number of aerosol samples analyzed for ROS activity (a total of 6), the data for both daytime and nighttime (i.e., interstitial) aerosols were combined together, and the following analyses were performed on the pooled data. Table S2 presents the results of the PCA conducted on the chemical components of daytime and interstitial aerosols. As can be seen in the table, two source factors were resolved, together explaining 95% of the variance in the data. The first source factor comprises species that most likely are of combustion origin (e.g., traffic, power plants), even though some of them (e.g., Fe, Mn, and Cu) may also come from non-exhaust traffic sources in other areas (Sanderson et al., 2014). The metals in this group, including V, Fe, Cu, Mn, and Ni, are considered toxic and known as redox-active metals (Argyropoulos et al., 2016; Sowlat et al., 2016; Wang et al., 2016). The second source factor consists of water-soluble components, including WSOC as well as the ionic components (methane-sulfonate, oxalate, sulfate, and nitrate) which are tracers of secondary aerosols (Hu et al., 2010; Hasheminassab et al., 2013). We should point out that the PCA





grouping of the chemical species reflects both the possible day-to-day variations in the contributions of the specific sources to PM mass as well as the diurnal cycles in concentrations governed by the specific fog scavenging rates (Figures S2, S3). Table 2 shows the Spearman rank correlation coefficients between the WS components and elemental/metallic species of the aerosols and the corresponding ROS levels. Very strong

positive correlations were observed between the ROS activity of aerosol samples and many of the WS components, which are also known tracers of secondary organic aerosols (SOA); for example, the Spearman rank correlation coefficients between the ROS activity and the concentrations of MSA and oxalate were 0.83 and 0.83, respectively. Similar as in the case of fog water, the observed association of inorganic ions and low-molecular weight organic acids with the aerosol ROS activity is probably because of the co-linearity of the

ionic species and (unspeciated) redox-active organic compounds. Interestingly, correlation coefficients are greater for the SOA tracers (oxalate, MSA) than for the bulk WSOC mixture (R = 0.66) and for the whole organic mass measured by the AMS (R = 0.43), suggesting that the ROS activity is prevalently contributed by the secondary fraction of particulate organic matter. However, Table 2 also shows strong positive and statistically significant correlations for several metal/elemental species, such as Fe, Ni, and Cu, which are

primary components of the aerosol. The correlation results were also corroborated by the outcome of the MLR analysis (Table 3), indicating that MSA (representative of the secondary organic aerosols source factor) and Ni (representative of the primary combustion-related source factor) were highly correlated with the ROS activity in the aerosol samples. Judging by the standardized coefficients (0.79 and 0.30 MicrogramZymosan/m$^3$ air for MSA and Fe, respectively) and partial correlation values (0.94 and 0.73 for

MSA and Fe respectively), the secondary organic aerosol fraction (represented by MSA in the MLR analysis) overall has a higher contribution to the aerosol ROS activity compared to combustion-related metals (represented by Fe in the MLR analysis). The results discussed above imply that during daytime, the aerosol ROS activity is driven by a combination of WS organics and redox active metals. However, during nighttime, when fog scavenging partitions the WS components into the water phase and enriches the nighttime (or

interstitial) aerosols with metals, the aerosol ROS activity is mainly driven by elemental/metallic components, while the ROS activity of fog is mostly driven by WS organic compounds.

WSOC is the main driver of toxicity in fog water and also a likely contributor of toxicity of PM1 during daytime when the contribution of WS components to toxicity is significant (Fig. 2). Submicron organic aerosols in rural Po Valley originate mainly from combustion processes (biomass burning) but a significant fraction of mass is

the product of chemical transformations occurring in wet aerosols and fog droplets ("aqueous SOA") (Gilardoni et al., 2016). Biomass burning organic tracers and their products of oligomerization formed in the aqueous phase contain hydroquinone and catechol moieties that can participate to redox reactions leading to ROS formation. The formation of such redox active organic compounds occurs in the aqueous phase, which explains a correlation with ionic tracers such as oxalate. Another source of reactive WSOC dissolved in

deliquesced aerosols and especially in fog water is the uptake of water-soluble reactive gases. It is well known that organic vapours, like low-molecular weight organic acids and carbonyls, can account for a significant fraction of WSOC in fog (up almost 50% according to Collett et al., (2008)). Some, like formic acid, exhibit ROS





activity (Du et al., 2008). In this study, the contribution of low-molecular weight organic acids to WSOC was on average 8% ± 4%, which represents a lower limit for the contribution of WS VOC to fog solutes (as formaldehyde and other carbonyls were not measured). The uptake of reactive gases can contribute to explain the excess of WSOC in fog water respect to the scavenged fraction from daytime PM1 (Figure 1 c,d)

and the higher intrinsic toxicity of fog components respect to PM1 (Figure 1 b).

## 4 Discussion and Conclusions

This study reports ROS activity data for PM1 and fog water samples at a continental rural area virtually free of local traffic emissions and where aerosol mass is prevalently contributed by biomass burning and
secondary organic and inorganic components (Gilardoni et al., 2014). The aerosol chemical composition differed substantially from that of urban PM for which the induced oxidative potential has been documented by several studies (Saffari et al., 2014) and can be put in relation to the redox-active activity of traffic-related aerosols (Ning and Sioutas, 2010). Despite the very limited in situ traffic emissions at the Po Valley site, the ROS levels recorded for PM (1879±851 and 1873±1199 MicroZymosan/mg during day and night respectively)
and fog samples (4142±1347 MicroZymosan/mg) in our study are comparable with those reported by Saffari et al. (2014), who reported ROS values of approximately 800 and 6000 MicrogramZymosan/mg PM in Los Angeles (LA) for PM2.5 and quasi-ultrafine PM (PM0.25), respectively. Additionally, in a more recent on-road study conducted in 2015 inside two major freeways of Los Angeles, the average (±SD) per-mass ROS activity of PM2.5 particles was found to be 3660 (±1743) and 3439 (±3058) MicrogramZymosan/mg PM for I-110 and
I-710, respectively (Shirmohammadi et al., 2016), indicating that the intrinsic toxicity of the Po Valley fog water is higher than that of PM2.5 particles collected on the freeways of Los Angeles.

Although direct comparison of the ROS activity of the fog and aerosol samples collected in this study with those of ambient aerosol previously reported by other studies is associated with some uncertainties (due mainly to the different size range of the particles collected and the fact that the ROS activity appears to be
strongly particle size-dependent (Sioutas et al., 2005; Saffari et al., 2014), nonetheless these results point to possible health effects associated with PM exposure during fog episodes in the Po Valley, the toxicity of which are comparable or, in many cases, higher than that of the highly toxic traffic-related PM. Our results show that the toxicity of aerosol particles accumulating in orographic depressions at the mid-latitudes during the cold season, which is normally peak concentration season for PM in many continental areas, can be further
amplified by the formation of fog, whose intrinsic toxicity is even greater than that of the original aerosol particles scavenged into the droplets. Moreover, the redox potential of fog solutes is mainly driven by oxidized organic compounds, which also explains the excess of ROS activity in fog water with respect to the scavenged fraction of the aerosol. The effect of secondary organic species on ROS activity in fog and aerosols in the Po Valley is clearly demonstrated by the fact that fog water exhibits a higher intrinsic toxicity with
respect to PM1, despite its depletion of redox active metals as a consequence of the systematically different




scavenging rates between WS and WI aerosol species. The contribution of WS secondary species to ROS activity is supported also by: a) the results of MLR analysis, b) the scavenging rate of ROS (71%) which is higher than that for most combustion-related metals (typically < 60%), corroborating the contribution of hygroscopic particles to ROS activity; and c) the results obtained from the filtered extracts indicating that ca.

50% of ROS activity is attributable to WS species in daytime (i.e., out-of-fog) conditions. The origin of secondary organic compounds responsible for the ROS activity of aerosol WSOC in the Po Valley cannot fully be elucidated based on this set of data. However, previous observations at the same site in the fog season showed that SOA are produced in large amounts by aqueous reactions in fog droplets and deliquesced aerosols starting from organic compounds emitted from biomass burning (Gilardoni et al., 2016). Production

of biomass burning SOA is accompanied by the formation of redox active organic compounds, including hydroquinones. We can therefore speculate that the enhancement of ROS activity in fog water is at least partly irreversible, as evaporating fog droplets become enriched on newly-formed redox-active secondary species. The relevance of secondary sources of toxicity in fog and fog-processed aerosols calls for more stringent controls on possible precursor emissions, which should be pursued by policy makers including

international authorities (as secondary PM components concentrations are often triggered by transboundary pollution) (Kiesewetter and Amann, 2014). Finally, the health effects of the exposure to fog water toxics depends on fog frequency, and in turn on climate conditions and climate change. Substantial reductions (-50%) of fog occurrence in the last 30 years have been documented for the Po Valley and many other European locations, and these dramatic changes have been attributed to global warming and changes in

cloud condensation nuclei concentrations (Vautard et al., 2009; Giulianelli et al., 2014). The enhanced toxicity of fog droplets observed in this study suggests that the historical reduction of fog frequency may result in an unintended improvement of air quality in many continental areas, overlapping also with the deliberate reduction of PM emissions put into practice since the early 90's in many developed countries.

**Acknowledgements**

This research was funded by Regione Emilia Romagna as part of the "Supersito" project (Decreto Regionale 428/10) and co-funded by the European Union's Seventh Framework Programme (FP7/2007-797 2013) under grant agreement no 603445 (BACCHUS). The authors would also like to acknowledge the financial support from the United States National Institute of Allergy and Infectious Diseases (award number:  5R01AI065617-15) and the National Institutes of Health (grant number: 5R01ES024936-02).

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





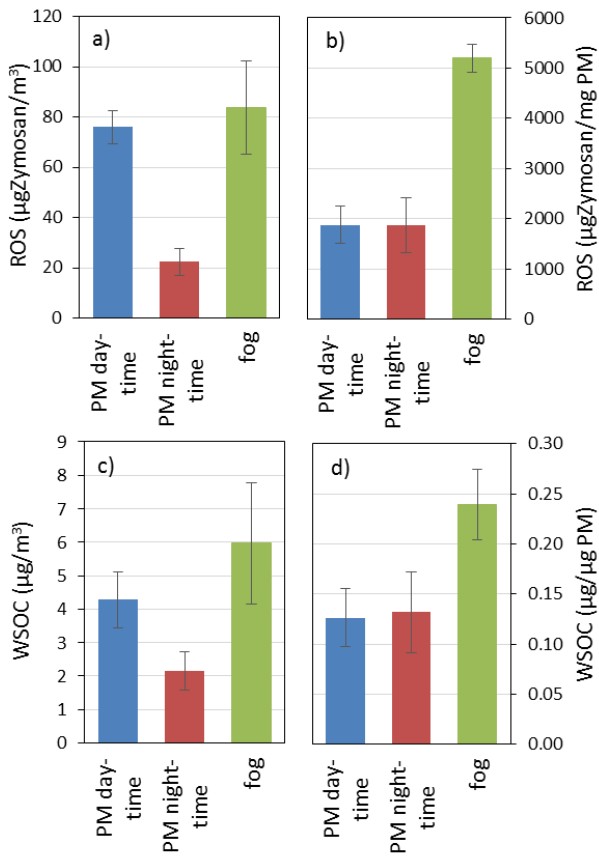

Figure 1: Per-volume (MicrogramZymosan/m³ air) (a) and per-mass (MicrogramZymosan/mg PM) (b) ROS
activity of the fog water and aerosol samples collected in the Po Valley in fall 2015. The results pertain to
parallel aerosol and fog samples.  The values for aerosol samples are based on the unfiltered ROS analysis
protocol. Bars represent geometric means and error bars correspond to one standard error (SD). Lower
panels (c and d): same as a) and b), but for WSOC.





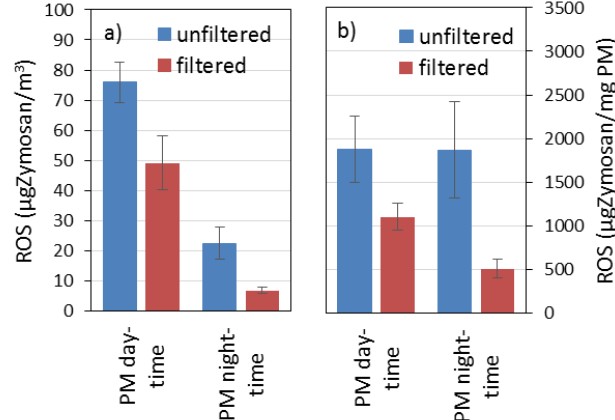

Figure 2: Per-volume (MicrogramZymosan/m³ air) (a) and per-mass (MicrogramZymosan/mg PM) (b) ROS activity of the aerosol samples collected in the Po Valley in fall 2015. The values for aerosol samples are based on both filtered (representing the redox activity of WS components) and unfiltered (representing the redox activity of both WS and WI components) ROS analysis protocols. Bars represent geometric means and error bars correspond to one standard error (SD).



Table 1 - Spearman rank correlation coefficients between per-volume concentrations of water-soluble species as well as metals/elements in the fog water samples and the corresponding ROS levels. Correlation coefficients which were statistically significant (at $P<0.05$) are highlighted in bold.

| Species | Corr Coeff | Species | Corr Coeff |
|---|---|---|---|
| **Water-soluble components** | | | |
| Acetate | 0.72* | **$Cl^-$** | **0.86** |
| Dimethylamine | 0.58 | **$K^+$** | **0.93** |
| **Ethylamine** | **0.90** | **$Mg^{2+}$** | **0.90** |
| Formate | 0.75* | **$Na^+$** | **0.89** |
| **Methansulfonate** | **0.89** | **$NH_4^+$** | **0.82** |
| Methylamine | 0.68* | $NO_2^-$ | -0.75 |
| Oxalate | 0.52 | **$NO_3^{2-}$** | **0.97** |
| **Trimethylamine** | **0.89** | $SO_4^{2-}$ | 0.61 |
| **$Ca^{2+}$** | **0.86** | **WSOC** | **0.93** |
| | | | |
| **Metals/Elements** | | | |
| Na | 0.26 | Ni | 0.26 |
| Mg | 0.60 | Cu | 0.49 |
| Al | 0.49 | Zn | 0.60 |
| Ca | 0.49 | As | 0.26 |
| Ti | 0.49 | Cd | 0.49 |
| V | 0.26 | Ba | 0.49 |
| Cr | 0.49 | Pb | 0.60 |
| Mn | 0.60 | Th | 0.49 |
| Fe | 0.49 | | |

*Denotes statistical significance at $P<0.1$.



Table 2 - Spearman rank correlation coefficients between concentrations of the metals/elements, markers of organic aerosol, and water-soluble (WS) components in the aerosol samples and the corresponding ROS levels. Statistically significant (P<0.05) correlation coefficients are highlighted in bold.

| Species | Corr Coeff | Species | Corr Coeff |
|---|---|---|---|
| **Markers of organic aerosol** | | | |
| m/z 44 (marker of OOA[1]) | 0.43 | m/z 60 (marker of BBOA[2]) | 0.31 |
| **Water-soluble components** | | | |
| Organic matter | 0.43 | $K^+$ | 0.60 |
| WSOC | 0.66 | $Mg^{2+}$ | 0.62 |
| Acetate | 0.66 | $Na^+$ | 0.60 |
| Formate | 0.66 | $NH_4^+$ | 0.31 |
| **Methansulfonate** | **0.83** | $NO_3^-$ | 0.66 |
| **Oxalate** | **0.83** | $SO_4^{2-}$ | 0.66 |
| $Cl^-$ | 0.66 | | |
| **Elements/metals** | | | |
| Na | 0.54 | Ni | 0.77* |
| Mg | 0.03 | Cu | 0.77* |
| Al | -0.25 | Zn | 0.60 |
| Ca | 0.33 | As | 0.66 |
| Ti | -0.52 | Cd | 0.31 |
| V | 0.74 | Ba | -0.15 |
| Cr | 0.26 | Pb | 0.66 |
| Mn | 0.66 | Th | 0.14 |
| Fe | 0.77* | | |

*Denotes statistical significance at P<0.1.

5     1 OOA: Oxygenated organic aerosol

2 BBOA: Biomass burning organic aerosol

Table 3 - Output of multiple linear regression (MLR) analysis using ROS activity as the dependent variable and ambient concentrations of the measured chemical species as independent variables.

| Category | Species | Standardized coefficients | Units | Partial R | R | P-value | $R^2$ |
|---|---|---|---|---|---|---|---|
| Fog Samples | WSOC | 0.97 | µg Zymosan/ $m^3$ air | 0.98 | 0.98 | <0.0001 | 0.96 |
| Aerosols | MSA | 0.79 | µg Zymosan/ $m^3$ air | 0.94 | 0.97 | 0.016 | 0.94 |
| | Fe | 0.30 | µg Zymosan/ $m^3$ air | 0.73 | | 0.161 | |