# Peer review of "Enhanced toxicity of aerosol in fog conditions in the Po Valley, Italy"

_Atmospheric Chemistry and Physics, 2017_

## Referee Comment (RC1) · Anonymous Referee #1 · 25 Feb 2017

This paper uses a rat macrophage assay to assess the toxicity of aerosols measured in the Po Valley during the cold season when fogs were often present and emissions from wood burning prevalent. The main finding is that under these conditions, fogs lead to SOA that is toxic (as per this specific assay used). The results of this paper are interesting in that they add to an existing body of literature showing the toxicity of aerosols increase with oxidation processes, along with the fact that wood smoke aerosol components have a high oxidative potential.

These authors have reported associations between their ROS measurement and WSOC in a number of past studies and asserted that the WSOC was secondary (for example, [Daher et al., 2012; Saffari et al., 2013; 2014]). The authors should note this and clearly state what is new about this work, ie, that the processing may be heterogeneous?

There is a substantial body of published literature on oxidative potential, albeit with different assays, that discusses the effect of oxidation on increased toxicity. Examples include chamber studies and analysis of ambient data (discussed more below). None of these, which are very pertinent to this paper, are cited in this work.

No evidence or reference is provided establishing that this assay (that is, this specific ROS measurement) is linked to adverse human health effects, although a health connection is implicitly assumed throughout. It would appear the implicit assumption is that because this is a cellular assay it can be directly connected to adverse human health responses, but there are many components to a cellular assay that can lead to various responses, so the connection is not established until empirically proven. This could be done by citing comparisons of their assay responses to other assays that have established links to health outcomes or oxidative stress markers or cite specific associations between this assay and health effects. As the paper stands, there really is no basis for asserting that these results specifically apply to human health, instead the author need to qualify this assertion throughout the paper.

Finally, there is the question of importance on a broader scale and associated assertions by the authors of wide ranging impacts. The authors suggest that populations are commonly exposed to aerosol that has been fog-processed, but is this true, what is the evidence for this? Quantitative support for this assumption should be provided to demonstrate that this mechanism is truly of broad importance, as stated. Overly expansive statements of the importance of this work should also be avoided throughout.

There are a number of other issues that also need to be addressed, which are discussed in more detail below.

Detailed comments.

Could not find any data on the various sample sizes (N).

The authors measured and report ROS of collected fog water and claim this is potentially linked to adverse health. How does this happen? Is the exposure route through inhaling fog drops? Likely not. Instead the argument is that the fog serves mainly as a chemical reactor that produces the toxic species. The drops evaporate and the fine PM is now more toxic. This assumes that all species in the fog contributing to ROS remain during evaporation, but it is stated that much of it is small molecular weight organic acids, which are likely very volatile and lost. If the fog ROS is equivalent to the ambient PM ROS, than these volatile species play no role. There seems to be some inconsistency in the author's arguments. Maybe this can be clarified.

A number of studies, such as chamber studies, have shown that if you take primary emissions, (say from a combustion source, like an automobile) and oxidize them, the oxidative potential substantially increases [Li et al., 2009] [McWhinney et al., 2011]. Likewise, chamber experiments in which SOA is produced from various VOCs show that some compounds, such as those found in biomass burning emissions, when oxidized have high intrinsic oxidative potentials [McWhinney et al., 2013]. It has even been shown that fresh soot that is subsequently oxidized has substantially increased oxidative potential (eg, [Antinolo et al., 2015; Shiraiwa et al., 2012]). All of these results are extremely pertinent to this work, but never cited nor discussed.

A variety of elemental concentrations of transition metals were measured, which are claimed to be redox active. Take Fe, for example. In the soluble form is redox active, but the insoluble form is not. Most measured elemental Fe is not soluble (many references show this) so no association to the water-soluble form, and hence redox activity, can be assumed a priori. The point here is that the use of elemental metal concentrations to infer toxicity through an oxidative stress response is not correct. This must be rectified in the manuscript.

The authors assert there are policy implications, but is it really a novel finding that aged biomass burning smoke is toxic? There are many publications on the toxicity of smoke to humans (some should have been cited). The main finding here is that cloud process increases the ROS produced by rat macrophage. This specific finding should be stated

in the context of overall known toxicity of smoke. (Ie, the authors could state something along the lines of, smoke is known to be toxic, here we show that fog processing of the smoke, increases the toxicity. . .).

Finally, the last line of the main text states: The enhanced toxicity of fog droplets observed in this study suggests that the historical reduction of fog frequency may result in an unintended improvement of air quality in many continental areas, overlapping also with the deliberate reduction of PM emissions put into practice since the early 90's in many developed countries This assumes that fogs are more effective than other atmospheric processes (eg, aqueous reactions in haze or gas phase oxidation followed by partitioning) in converting wood smoke emissions to species toxic to humans. Is there any evidence for this? The point is what proof do the authors have that if the fogs were not present the aerosol would not still chemically evolve over time to a similar toxicity as fog-processed smoke?

Refs: Antinolo, M., M. D. Willis, S. Zhou, and J. P. D. Abbatt (2015), Connecting the oxidation of soot to its redox cycling abilities, Nature Comm, 6:6812 DOI: 10.1038/ncomms7812

Daher, N., A. Ruprecht, G. Invernizzi, C. D. Marco, J. Miller-Schulze, J. B. Heo, M. M. Shafer, B. R. Shelton, J. J. Schauer, and C. Sioutas (2012), Characterization, sources and redox activity of fine and coarse particulate matter in Milan, Italy, Atmos. Env.(49), 130-141.

Li, Q., A. Wyatt, and R. M. Kamens (2009), Oxidant generation and toxicity enhancement of aged-diesel exhaust, Atm. Envir., 43, 1037-1042.

McWhinney, R. D., S. Zhou, and J. P. D. Abbatt (2013), Naphthalene SOA: redox activity and naphthoquinone gas-particle partitioning, Atm. Chem. Phys. , 13, 9731-9744.

McWhinney, R. D., S. S. Gao, S. Zhou, and J. P. D. Abbatt (2011), Evaluation of the Effects of Ozone Oxidation on Redox-Cycling Activity of Two-Stroke Engine Exhaust

Particles, Environ. Sci. Technol., 45(6), 2131-2136.

Saffari, A., N. Daher, M. M. Shafer, J. J. Schauer, and C. Sioutas (2013), Seasonal and spatial variation in reactive oxygen species activity of quasi-ultrafine particles (PM0.25) in the Los Angeles Basin and its association with chemical species, Atm. Envir., 79, 566-575.

Saffari, A., N. Daher, M. M. Shafer, J. J. Schauer, and C. Sioutas (2014), Global persective on the oxidative potential of airborne particulate matter: A synthesis of research findings, Environ. Sci. Technol.

Shiraiwa, M., K. Selzle, and U. Poschl (2012), Hazardous components and health effects of atmospheric aerosol particles: reactive oxygen species, soot, polycyclic aromatic compounds and allergenic proteins, Free Radic. Res., 46(8), 927-939.

---

## Author Comment (AC1) · 2 Mar 2017

We thank the Reviewer for the constructive criticisms. Below we provide a point-by-point rebuttal to his/her specific comments. However, his/her remarks about the lack of recognition of past studies do not hold, as many of the papers suggested by the Referee have already been included in the references list of the manuscript, and clearly the Reviewer simply missed them.

*These authors have reported associations between their ROS measurement and WSOC in a number of past studies and asserted that the WSOC was secondary (for example, [Daher et al., 2012; Saffari et al., 2013; 2014]). The authors should note this and clearly state what is new about this work, ie, that the processing may be heterogeneous?*

**REPLY**: The statistical association between ROS activity (cellular essay) and WSOC concentrations has already been investigated at a relatively small number of sites (see the review by Saffari et al. [Environ. Sci. Technol. 2014, 48, 7576–7583]). The originality of the present study relies on two aspects:

1. As the Referee pointed out, we investigate here the effect of heterogeneous processes (fog formation) on ROS activity in the particles.
2. Past experiments were based on aerosol samples in which WSOC and other redox-active agents (including metals) were actually physically mixed together, and their contributions to ROS activity could be disentangled only with statistical tools. Here instead, we provide observations of ROS activity of bulk aerosol particles and for interstitial aerosols, with the latter characterized by being naturally depleted of soluble compounds as a consequence of fog scavenging. Conversely, fog water samples largely recover WSOC of the original particle population while the insoluble species are mostly left out in the interstitial aerosol phase. We believe that the exploitation of the natural partitioning effect of fog on aerosol species exhibiting different water solubilities is an original contribution of our study to the research on the nature of the chemical species governing redox activity in the aerosol.

*There is a substantial body of published literature on oxidative potential, albeit with different assays, that discusses the effect of oxidation on increased toxicity. Examples include chamber studies and analysis of ambient data (discussed more below). None of these, which are very pertinent to this paper, are cited in this work.*

**REPLY**: The references suggested by the reviewer have already been included in the manuscript. See the detailed comment below.

*No evidence or reference is provided establishing that this assay (that is, this specific ROS measurement) is linked to adverse human health effects, although a health connection is implicitly assumed throughout. It would appear the implicit assumption is that because this is a cellular assay it can be directly connected to adverse human health responses, but there are many components to a cellular assay that can lead to various responses, so the connection is not established until empirically proven. This could be done by citing comparisons of their assay responses to other assays that have established links to health outcomes or oxidative stress markers or cite specific associations between this assay and health effects. As the paper stands, there really is no basis for asserting that these results specifically apply to human health, instead the author need to qualify this assertion throughout the paper.*

**REPLY**: The association between ROS activity and adverse health outcomes is certainly a subject of ongoing investigation. Nevertheless, it is not true that cellular ROS assays have not been connected to adverse human

health responses, and it is unfair to state that "*As the paper stands, there really is no basis for asserting that these results specifically apply to human health*", because the link between ROS assays and human health effects has already been established in few epidemiological studies, which have already been explicitly cited in the paper (Page 4, Line 28).

*Finally, there is the question of importance on a broader scale and associated assertions by the authors of wide ranging impacts. The authors suggest that populations are commonly exposed to aerosol that has been fog-processed, but is this true, what is the evidence for this? Quantitative support for this assumption should be provided to demonstrate that this mechanism is truly of broad importance, as stated. Overly expansive statements of the importance of this work should also be avoided throughout.*

**REPLY**: Fog and low-level clouds are transient phenomena in the atmosphere but their occurrence can in fact be very high in certain areas of the globe. This is especially true for highly-populated regions in orographic basins in wintertime. Cermak et al. (2009) showed that several pollution hotspots in Europe, including Benelux, the Ruhr district, the basins of Paris and London and the Po Valley, experience low-level clouds and fogs for 35% to 60% of the days in winter months. The fraction of fog days in fall/winter in the Californian Central Valley (6.5 millions inhabitants) is ca. 20% according to Baldocchi and Waller (2014). Fog frequencies of ca. 10% in winter are also characteristic of the Yangtze River corridor (Niu et al., 2010), and even greater values (20% to more than 35%) are typical of the Indo-Gangetic plain (Saraf et al., 2011). All these regions of the globe commonly experience PM pollution peaks in winter months, during the fog season. In this season of the year, the same stable weather conditions favor the accumulation of air pollutants and fog formation. Therefore, fog-processing is potentially a major driver for secondary aerosol formation in wintertime at all these sites. In Gilardoni et al. (2016), we provided a first estimate of SOA produced by aqueous-phase processing of smoke particles in Europe: 0.1 to 0.5 Tg of organic carbon per year, corresponding to $4-20\%$ of total primary OA emissions in the region.

*Could not find any data on the various sample sizes (N).*

**REPLY**: The information can be found on page 2 Lines 33-35, and Page 3 Lines 1-5:

"Fog samples were collected from 30 November to 30 December 2015 at the meteorological station Giorgio Fea in San Pietro Capofiume (44°39'15" latitude, 11°37'29" longitude), a rural site located 30 km northeast of Bologna (Italy) in the eastern part of the Po Valley (northern Italy). From 30 November to 4 December, an intensive observation period was scheduled, with the concurrent sampling of fog and aerosol samples and the deployment of a HR-ToF-AMS (Aerodyne Research) for online aerosol measurements. During the sampling campaign, a total of 6 aerosol samples and 16 fog samples were collected. Additionally, fog samples collected after the intensive observation period (after 4 December) were pooled in groups of two or three for the analysis of metals and Oxidative Potential. "

We further specify that the full chemical analyses (WSOC, ion chromatography, metals) was performed on 6 aerosol samples and 7 fog samples. ROS activity analysis was carried out on 20 samples (= 6 aerosol + 7 unfiltered fog + 7 filtered fog samples). Finally, sample size information is specified in every table in the present version of the manuscript.

*The authors measured and report ROS of collected fog water and claim this is potentially linked to adverse health. How does this happen? Is the exposure route through inhaling fog drops? Likely not. Instead the argument is that the fog serves mainly as a chemical reactor that produces the toxic species. The drops*

*evaporate and the fine PM is now more toxic. This assumes that all species in the fog contributing to ROS remain during evaporation, but it is stated that much of it is small molecular weight organic acids, which are likely very volatile and lost. If the fog ROS is equivalent to the ambient PM ROS, than these volatile species play no role. There seems to be some inconsistency in the author's arguments. Maybe this can be clarified.*

**REPLY**: The Referee is right in pointing out that the volatility characteristics of redox-active WSOC in fog water eventually affect exposure. If we hypothesize that these are truly absorbed VOCs, such as formic acid, than their partitioning is completely reversible. However, the Referee assumes that fog droplets dry out once inhaled. On the contrary, fog droplets are expected to travel along the respiratory tract, just alike the droplets of nebulized solutions produced by aerosol generators for clinical use. Airways are much warmer with respect to ambient air but also humid close to saturation, and water vapor diffuses quicker than temperature (a physical process which is also at the basis of cloud condensation nuclei counters), therefore evaporation of fog droplets along their travel in the respiratory tract can be much reduced. The size range of Po Valley fog droplets, spanning between 3 and 30 µm (Heintzenberg et al., 1998) indicates that the inhaled fraction can deposit all over the respiratory tract down to the lungs. VOC evaporating from drying droplets can be exhaled but also become adsorbed to the wet tissues of the airways. The issue of the deposited fraction of the volatile fraction of fog solutes is complex and cannot be fully assessed in this paper. It is worthwhile to note, however, that non-volatile SOA are also expected to form from aqueous oxidation of water-soluble VOC and SVOC. For instance, evidence for dimerization of phenolic compounds in Po Valley fog was presented by Gilardoni et al. (2016). Such SOA compounds exhibit a quinoid structure and are potentially redox-active. The correlation found in this study between ROS activity and oxalic acid concentration (a tracer for highly oxidized, low-volatility aqueous SOA) supports the hypothesis of redox-active WSOC components of reduced volatility.

Finally, in respect to the Referee's comment "If the fog ROS is equivalent to the ambient PM ROS, than these volatile species play  no role", we would like to specify that, as discussed in the paper, on a per-volume basis, daytime aerosols and fog exhibit comparable ROS, but this is only due to the large mass concentration of daytime aerosols. However, on a per-mass basis, which is the relevant metric here in discussing toxicity, ROS activity of fog is 2.5 times larger than that of both daytime and nighttime aerosols. This is clearly shown in the most important figure of the paper (Figure 1). So there is actually no inconsistency in the results and the statements made in the manuscript.

*A number of studies, such as chamber studies, have shown that if you take primary emissions, (say from a combustion source, like an automobile) and oxidize them, the oxidative potential substantially increases [Li et al., 2009] [McWhinney et al., 2011]. Likewise, chamber experiments in which SOA is produced from various VOCs show that some compounds, such as those found in biomass burning emissions, when oxidized have high intrinsic oxidative potentials [McWhinney et al., 2013]. It has even been shown that fresh soot that is subsequently oxidized has substantially increased oxidative potential (eg, [Antinolo et al., 2015; Shiraiwa et al., 2012]). All of these results are extremely pertinent to this work, but never cited nor discussed.*

**REPLY**: On the contrary, several of these studies have already been cited in the manuscript (Page 2, Lines 9-13).

*A variety of elemental concentrations of transition metals were measured, which are claimed to be redox active. Take Fe, for example. In the soluble form is redox active, but the insoluble form is not. Most measured elemental Fe is not soluble (many references show this) so no association to the water-soluble form, and hence redox activity, can be assumed a priori. The point here is that the use of elemental metal concentrations to infer toxicity through an oxidative stress response is not correct. This must be rectified in the manuscript.*

**REPLY**: Our observations indicate a specific contribution of metals to ROS activity of PM distinct from that of SOA (Table 3). At the same time, filtration experiments showed that ROS activity is contributed by both soluble and insoluble components of the aerosol (Figure 2). Since several metals showed an enrichment in interstitial particles (like Cr, Mn, Fe, Cd and Pb, Figure S3b) which are depleted of soluble compounds, we explained the ROS activity of the unfiltered extracts of interstitial aerosol samples (Figure 2) with the presence of water-insoluble transition metals. This does not exclude a contribution of metals also to the soluble fraction of redox-active particulate matter (which, for daytime aerosols, can be large). Contrary to the Referee's indications, there is evidence that the soluble fraction of metals such as V, Zn, As and Cd can be > 50% in submicron aerosol samples (PM2.5) (Heal et al.; 2005). Past experiments dealing with fog chemistry in the Po Valley (Mancinelli et al., 2005) highlighted large soluble fractions for several transition metals (e.g., 77% and 81% for Zn and Cu, respectively). The same study showed that, even if Fe is preferentially distributed to the insoluble core of fog droplets, the soluble fraction still accounts for more than 1/3 of total elemental iron. Therefore, a contribution to ROS activity from partially-soluble transition metals such as Fe cannot be ruled out and, on the contrary, is expected.

*The authors assert there are policy implications, but is it really a novel finding that aged biomass burning smoke is toxic? There are many publications on the toxicity of smoke to humans (some should have been cited). The main finding here is that cloud process increases the ROS produced by rat macrophage. This specific finding should be stated in the context of overall known toxicity of smoke. (Ie, the authors could state something along the lines of, smoke is known to be toxic, here we show that fog processing of the smoke, increases the toxicity...).*

**REPLY**: Although we agree with the Reviewer that there have been many studies indicating the toxicity of biomass burning particles, the main point of the paper and its novelty is the resulting increased toxicity from fog processing, and not the toxicity of biomass burning aerosol itself. This is clearly a novel finding which has not been reported previously. Nonetheless, we accept the Reviewer's comment and we will cite relevant studies on the toxicity of smoke particles to humans.

*Finally, the last line of the main text states: The enhanced toxicity of fog droplets observed in this study suggests that the historical reduction of fog frequency may result in an unintended improvement of air quality in many continental areas, overlapping also with the deliberate reduction of PM emissions put into practice since the early 90's in many developed countries This assumes that fogs are more effective than other atmospheric processes (eg, aqueous reactions in haze or gas phase oxidation followed by partitioning) in converting wood smoke emissions to species toxic to humans. Is there any evidence for this? The point is what proof do the authors have that if the fogs were not present the aerosol would not still chemically evolve over time to a similar toxicity as fog-processed smoke?*

**REPLY**: This Referee's comment is highly speculative. Our conclusion statement in the paper is a direct implication of our own findings. We do not claim that the in-fog processes are the most efficient processes governing ROS activity in the Po Valley aerosol, but, since fog frequency has decreased with time – as it is documented by visibility and liquid water content data –, the resulting specific impact is a parallel decrease in redox-active species concentrations in the particles. We are not aware if in the meantime something has changed also in respect to all other possible processes involving redox-active compound formation in SOA. We simply have no data for this hypothesis, nor Referee has provided data in support of it.

**References:**

Baldocchi, D., and Waller, R. (2014), Winter fog is decreasing in the fruit growing region of the Central Valley of California, Geophys. Res. Lett., 10.1002/2014GL060018.

Cermak, J., R. M. Eastman, J. Bendix, S. G. Warren (2009), European climatology of fog and low stratus based on geostationary satellite observations, Q. J. R. Meteorol. Soc., 135, 2125–2130.

Heal, M. R., Hibbs, L. R., Agius, R. M., & Beverland, I. J. (2005). Total and water-soluble trace metal content of urban background PM 10, PM 2.5 and black smoke in Edinburgh, UK. Atmos. Environ., 39(8), 1417-1430.

Heintzenberg, J., M. Wendisch, B. Yuskiewicz, D. Orsini, A. Wiedensohler, F. Stratmann, G. Frank, B. G. Martinsson, D. Schell, S. Fuzzi, G. Orsi (1998), Characteristics of haze, mist and fog, Contr. Atmos. Phys., 71, 21 -31.

Gilardoni, S., Massoli, P., Paglione, M., Giulianelli, L., Carbone, C., Rinaldi, M., Decesari, S., Sandrini, S., Costabile, F., Gobbi, G. P., Pietrogrande, M. C., Visentin, M., Scotto, F., Fuzzi, S., and Facchini, M. C. (2016): Direct observation of aqueous secondary organic aerosol from biomass-burning emissions, PNAS, 113, 10013-10018, 10.1073/pnas.1602212113.

Niu, F., Z. Li, C. Li, K.-H. Lee, M. Wang (2010), Increase of wintertime fog in China: Potential impacts of weakening of the Eastern Asian monsoon circulation and increasing aerosol loading, J. Geophys. Res. 115, doi:10.1029/2009JD013484.

Mancinelli, V., S. Decesari, M. C. Facchini, S. Fuzzi, F. Mangani, Partitioning of metals between the aqueous phase and suspended insoluble material in fog droplets (2005), Ann Chim., 95, 275-290.

Saraf, A. K.., A. K. Bora, J. Das, V. Rawat, K. Sharma, S. K. Jain (2011), Winter fog over the Indo-Gangetic Plains: mapping and modelling using remote sensing and GIS, Nat. Hazards, 58, 199–220.

---

## Referee Comment (RC2) · Anonymous Referee #2 · 26 Mar 2017

Decesari et al. present a very interesting and novel analysis of Reactive Oxygen Species (ROS), including examinations of ROS in aerosol and fog and the effects of fog scavenging and chemistry on aerosol ROS. The design of the experiment, which takes advantage of differential fog scavenging of different aerosol ROS components (e.g., metals vs. WOC), is particularly clever. The manuscript is well written, concise and even spare in its style, effectively conveying a lot of information in a compact form. I have a few, mostly minor, comments that should be addressed:

1. In performing mass balances of ROS and other components across aerosol and fog over time, the authors are assuming that there are no significant changes due to factors such as changes in boundary layer depth and fog drop deposition. Nocturnal cooling of the top of a fog layer typically leads to entrainment of air from above the boundary layer and associated growth in boundary layer depth. The entrained air is likely to have very

different composition than the air originally in the boundary layer. This entrainment of air of differing composition can alter the mass balance. Likewise, significant deposition of fog water over the course of an episode can substantially cleanse the boundary layer of scavenged particles, again altering the mass balance. These factors need to be clearly outlined as sources of possible error in the mass balance analysis that is central to the paper.

2. Lines 9-12 of abstract: this sentence should be rewritten to more clearly distinguish primary and secondary particle source contributions that the authors are referring to.

3. Section 2.2: more information should be presented concerning cleaning of the fog sampler and any contamination contained in collector blanks.

4. P3, line 12: This should be the PVM-100 not PVM-10.

5. P.3, lines 15-16: I suggest you explain to the reader that multiplying by LWC yields "air equivalent concentrations"

6. Section 2.3: Please explain to the reader why you chose to filter the fog samples. Suspended particulate matter inside fog drops is also scavenged material that should be considered as part of the overall system ROS mass balance.

7. P.3, line 25: change "chromatographers" to "chromatographs"

8. P. 4, lines 9-10: I am confused why the authors would extract fog water with Milli-Q water. Is this statement in error?

9. P. 5, line 29: change "what observed" to "what is observed"

10. P. 5, line 30: change "ammonia" to "ammonium" since you are discussing ionic species here

11. P. 6, line 15: change "adsorbed" to "absorbed"

12. P. 6, lines 35-36: The statement "fog scavenging denudes particles of WS components" is misleading. It sounds like material is being stripped off particles while what is really happening is that the fog is selectively scavenging some particles and leaving others intact in the aerosol. Please rephrase.

13. P. 9, lines 3-5: This sentence needs to be rewritten to improve the grammar.

―――――――――――――――――――――

---

## Referee Comment (RC3) · Anonymous Referee #3 · 2 Apr 2017

The paper from Decesari et al., measures the capability of aerosols to generate reactive oxygen species during the fog conditions in Po, Valley, Italy. The authors use a suitable assay, which is biologically relevant to indirectly interpret the toxicity of ambient aerosols. The study is well conducted although the sample size is small (n∼6). However, the reviewer can understand the intricacies involved in collecting the samples with enough mass for conducting all the toxicity and chemical analysis, particularly in the ambient conditions as in the study. The measurements are novel and this is probably the first time, ROS activity of the fog has been measured. The manuscript is very well written and the results are interesting and highly useful from the perspective of policy intervention, particularly in controlling the trans boundary movement of the pollutants. I support the publication of this manuscript. However, I have few comments below,

which can help the authors to further improve upon their work.

Page 1, Line 34: There have been many epidemiological evidences showing the links between traffic pollution and adverse health effects (e.g. Brunekreef et al., Journal of Exposure Science and Environmental Epidemiology (2007) 17, S61–S65; doi:10.1038/sj.jes.7500628). Janssen et al., 2011 is not the first evidence.

Page 2, Line 1: Bates and Fang et al., 2015 were not the toxicological studies. Bates et al., 2015 should be considered as epidemiological study and Fang et al., 2015 didn't report any direct linkages with the health impacts.

Page 3, Line 15-16: The conversion of $\mu$g/mL to $\mu$g /m3 needs more description. What was the sampling flow rate, etc?

Page 4, Line 14: The unfiltered fraction was directly assayed for the ROS activity but this fraction would be containing a lot of quartz fibers as well. These fibers are also shown to be toxic to the cells. How did the authors make sure that their results are well controlled in this environment?

Page 4, Line 22-23: What was the level of blank? And how much was the typical response from the sample in comparison to blanks?

Page 6, Line 34: The secondary ionic species. . .. like what? Are these secondary ionic species shown to be toxic or ROS- active?

Page 7, Line 14-16: Although, it seems possible that the high correlation of inorganic species such as SO4-2, NO3-, etc. could be due to their co-linearity with the WSOC, however, recent studies (Environmental Science & Technology 51 (5), 2611-2620; Environ Sci Technol. 2012, 46(12):6637-44) have shown that SO4-2 enhanced acidity of the aerosol can solubilize the metals (such as Fe and Cu), which are known to generate ROS. Do the authors think to include such possibility in their work?

Page 8, Line 14: See my earlier comment, could the toxicity of Fe, Ni and Cu be due SO4-2 enhanced acidity and the solubility of these metals?

Page 9, Line 4: There should be "with" between "water" and "respect"

Page 10, Line 2: What do you mean by scavenging rate of ROS? I think this sentence needs to be either further cleared or modified.

---

## Author Comment (AC2) · 10 Apr 2017

**Reply (in green) to REFEREE 2**

Decesari et al. present a very interesting and novel analysis of Reactive Oxygen Species (ROS), including examinations of ROS in aerosol and fog and the effects of fog scavenging and chemistry on aerosol ROS. The design of the experiment, which takes advantage of differential fog scavenging of different aerosol ROS components (e.g., metals vs. WOC), is particularly clever. The manuscript is well written, concise and even spare in its style, effectively conveying a lot of information in a compact form.

Reply: we thank the Referee for her/his comments, which turn helpful in clarifying specific points of the manuscript.

I have a few, mostly minor, comments that should be addressed:

1. In performing mass balances of ROS and other components across aerosol and fog over time, the authors are assuming that there are no significant changes due to factors such as changes in boundary layer depth and fog drop deposition. Nocturnal cooling of the top of a fog layer typically leads to entrainment of air from above the boundary layer and associated growth in boundary layer depth. The entrained air is likely to have very different composition than the air originally in the boundary layer. This entrainment of air of differing composition can alter the mass balance. Likewise, significant deposition of fog water over the course of an episode can substantially cleanse the boundary layer of scavenged particles, again altering the mass balance. These factors need to be clearly outlined as sources of possible error in the mass balance analysis that is central to the paper.

Reply: The Referee is right. Nocturnal cooling at top of the fog layer makes this unstable, as documented by previous observations at the site (Wobrock et al., 1998). Occult depositions of fog were also investigated in past campaigns (Jaeschke et al., 1998). In general, the real atmosphere is not a closed system in any dimensions, even horizontally. Therefore, our interpretation of the results, assuming a "box model" where sinks and external sources of materials are ignored, can be valid only in a first approximation. Our assumption is primarily supported by the mass balance achieved between daytime PM concentrations and the lumped nighttime PM and fog components concentrations (Fig. S1): the mass balance shows that PM behaved approximately conservatively in the PBL in the days considered in this study. We acknowledge, however, that the PM mass balance can be affected by compensating fluxes of particles in and out the PBL (entrainments and depositions).

2. Lines 9-12 of abstract: this sentence should be rewritten to more clearly distinguish primary and secondary particle source contributions that the authors are referring to.

Reply: Estimates of the relative contributions of biomass burning and secondary sources to PM mass concentrations during the November 2015 fog campaign are currently not available. Nevertheless, the AMS proxies for biomass burning POA and for oxidized organic aerosols (m/z 60 and m/z 44, in Table 2) indicates that the organic composition during the November 2015 field campaign is compatible with that observed during the previous experiments at the site where quantitative source apportionment of submicron PM was carried out (Gilardoni et al., 2016). This is why we stated that "Po Valley PM, which is mainly composed of biomass burning and secondary aerosols" in the abstract, without reporting the actual shares of PM mass. The point here is that the PM composition in a rural area like the Po Valley is *qualitatively* very different from that of urban environments, much more impacted by traffic emissions and more extensively studies for ROS.

3. Section 2.2: more information should be presented concerning cleaning of the fog sampler and any contamination contained in collector blanks.

Reply: The collector and its strings were extensively cleaned at the beginning of the fog season (approximately two weeks before the samples object of this study were collected). In particular, the strings (stainless steel) were carried to a chemical laboratory where they were gently brushed to remove any stuck residues from fog sampling in the previous winter and washed with milliQ water in an ultrasonic bath. Finally, they were brought back to the field and mounted on the fog collector before first the automatic sampling system was switched on around mid-November. This information will be added to Section 2.2.

4. P3, line 12: This should be the PVM-100 not PVM-10.

Reply: correct. It is PVM-100.

5. P.3, lines 15-16: I suggest you explain to the reader that multiplying by LWC yields "air equivalent concentrations"

Reply: we will add the following sentence to the text: "Concentrations of analytes in fog samples, expressed in $\mu g\ mL^{-1}$, were converted into $\mu g\ m^{-3}$ by multiplying with the fog liquid volume ($mL\ m^{-3}$). The latter was not estimated by the mass of sampled fog water multiplied by the flux of the fog collector, because the collection efficiency is typically much smaller than 1 (about 40%, according to Fuzzi et al. 1997). We used instead the liquid water content (LWC) measured by the PVM-100 and averaged over the sampling time of the fog collector to multiply by the concentrations of chemicals in fog water to get air equivalent concentrations ($\mu g\ m^{-3}$) of the fog components".

6. Section 2.3: Please explain to the reader why you chose to filter the fog samples. Suspended particulate matter inside fog drops is also scavenged material that should be considered as part of the overall system ROS mass balance.

Reply: true, and in fact we performed ROS analyses on both filtered and unfiltered extracts. Since fog chemical components partition selectively between the soluble and insoluble phases, performing the analysis of ROS activity with and without filtration provides useful information on the nature of the redox-active compounds (Fig. 2).

7. P.3, line 25: change "chromatographers" to "chromatographs"

Reply: will be corrected.

8. P. 4, lines 9-10: I am confused why the authors would extract fog water with Milli-Q water. Is this statement in error?

Reply: This was mentioned in error. Please see below the corrected sentence: "In this method, samples (filters) were first extracted using an initial sonication period for 15 min with high-purity 10 Milli-Q (18 mΩ) water."

9. P. 5, line 29: change "what observed" to "what is observed"

Reply: will be corrected.

10. P. 5, line 30: change "ammonia" to "ammonium" since you are discussing ionic species here

Reply: will be corrected.

11. P. 6, line 15: change "adsorbed" to "absorbed"

Reply: accepted.

12. P. 6, lines 35-36: The statement "fog scavenging denudes particles of WS components" is misleading. It sounds like material is being stripped off particles while what is really happening is

that the fog is selectively scavenging some particles and leaving others intact in the aerosol. Please rephrase.

Reply: Accepted. The sentence will be changed into "as fog scavenges the WS components of the particles".

13. P. 9, lines 3-5: This sentence needs to be rewritten to improve the grammar.

Reply: the text will be rephrased as follows: "The uptake of reactive gases could  explain the excess of WSOC in fog water compared to the scavenged fraction of daytime PM1 (Figure 1 c,d)  as well as the higher intrinsic toxicity of fog components , as shown in Figure 1 b".

REFERENCES:

Fuzzi, S., Orsi, G., Bonforte, G., Zardini, B., and Franchini, P. L.: An automated fog water collector suitable for deposition networks: Design, operation and field tests, Water Air and Soil Pollution, 93, 383-394, 10.1023/a:1022108801083, 1997.

Gilardoni, S., Massoli, P., Paglione, M., Giulianelli, L., Carbone, C., Rinaldi, M., Decesari, S., Sandrini, S., 10 Costabile, F., Gobbi, G. P., Pietrogrande, M. C., Visentin, M., Scotto, F., Fuzzi, S., and Facchini, M. C.: Direct observation of aqueous secondary organic aerosol from biomass-burning emissions, Proceedings of the National Academy of Sciences of the United States of America, 113, 10013-10018, 10.1073/pnas.1602212113, 2016.

Jaeschke, W., Dierssen, J. P., Guenther, A., Schickedanz, U., Wolf, A., Ricci, L., Arends, B. G., Mass fluxes and chemical pathways during a fog event, Contr. Atmos. Phys., 71, 1, 145 – 157, 1998.

Wobrock, W., Jaeschke, W., Schell, D., Teichmann, U., Wendisch, M., Mertes, S., Laubach, J., Fuzzi, S., Orsi, G., Observations of the turbulence structure of wind, temperature and liquid water content in a foggy surface layer, Contr. Atmos. Phys., 71, 1, 171 – 187, 1998.

---

## Author Comment (AC3) · 10 Apr 2017

**Reply (in green) to REFEREE 3**

The paper from Decesari et al., measures the capability of aerosols to generate reactive oxygen species during the fog conditions in Po, Valley, Italy. The authors use a suitable assay, which is biologically relevant to indirectly interpret the toxicity of ambient aerosols. The study is well conducted although the sample size is small (n ~ 6). However, the reviewer can understand the intricacies involved in collecting the samples with enough mass for conducting all the toxicity and chemical analysis, particularly in the ambient conditions as in the study. The measurements are novel and this is probably the first time, ROS activity of the fog has been measured. The manuscript is very well written and the results are interesting and highly useful from the perspective of policy intervention, particularly in controlling the trans boundary movement of the pollutants. I support the publication of this manuscript. However, I have few comments below, which can help the authors to further improve upon their work.

Reply: we thank the Referee for the useful comments. We provide a point-by-point reply below.

Page 1, Line 34: There have been many epidemiological evidences showing the links between traffic pollution and adverse health effects (e.g. Brunekreef et al., Journal of Exposure Science and Environmental Epidemiology (2007) 17, S61–S65; doi:10.1038/sj.jes.7500628). Janssen et al., 2011 is not the first evidence.

Reply: correct, however, most epidemiological studies used $NO_2$ concentrations to assess the effect of differential exposures to traffic pollution within cities on human health, while Janssen et al. (2011) was among the first studies employing black carbon as a tracer for primary PM.

Page 2, Line 1: Bates and Fang et al., 2015 were not the toxicological studies. Bates et al., 2015 should be considered as epidemiological study and Fang et al., 2015 didn't report any direct linkages with the health impacts.

Reply: The Referee is right about Bates et al. (2015). Its quotation in the text will be transferred to Page 1, line 34 among the citations of other epidemiological studies. The study of Fang et al., (2015) can instead be considered a toxicological study, as it makes use of a chemical assay for ROS activity (DTT assay).

Page 3, Line 15-16: The conversion of µg/mL to µg/m3 needs more description. What was the sampling flow rate, etc?

Reply: we will add the following text: "Concentrations of analytes in fog samples, expressed in $\mu g\ mL^{-1}$, were converted into $\mu g\ m^{-3}$ by multiplying with the fog liquid volume ($mL\ m^{-3}$). The latter was not estimated by the mass of sampled fog water multiplied by the flux of the fog collector, because the collection efficiency is typically much smaller than 1 (about 40%, according to Fuzzi et al. 1997). We used instead the liquid water content (LWC) measured by the PVM-100 and averaged over the sampling time of the fog collector to multiply by the concentrations of chemicals in fog water to get air equivalent concentrations ($\mu g\ m^{-3}$) of the fog components".

Page 4, Line 14: The unfiltered fraction was directly assayed for the ROS activity but this fraction would be containing a lot of quartz fibers as well. These fibers are also shown to be toxic to the cells. How did the authors make sure that their results are well controlled in this environment?

Reply: As explained in Section 2.5, the effect of quartz fibers from the filters is accounted for by the analysis of the blanks. Unfiltered extracts of blank filters (n = 2) exhibit a ROS activity of $280 \pm 13$ Fluorescence Units (FU, measured by the microplate reader), which, upon filtration, decreases to $27 \pm 23$ FU. These analyses confirm that the contribution of filter fibers to ROS activity is substantial, in agreement with the Referee's comments. Nevertheless, the ROS activity measured for the samples is much greater than the blank levels (levels as high as $3896 \pm 350$ FU for the not-filtered and $1518 \pm 118$ FU for the filtered sample fractions

were measured, respectively). The filters were in fact highly loaded, as they originated from sampling for 12h with a HiVol system in a polluted atmosphere (PM1 ~ 35 µg/m$^3$ in daytime, Fig. S1).

Page 4, Line 22-23: What was the level of blank? And how much was the typical response from the sample in comparison to blanks?

Reply: see reply to previous comment.

Page 6, Line 34: The secondary ionic species .... like what? Are these secondary ionic species shown to be toxic or ROS- active?

Reply: the reasoning in this part of the text was based solely on similarities in the scavenging behavior of redox-active compounds (measured by ROS activity) with that of all the chemical species analyzed in daytime and night-time PM samples. The actual association of redox-active compounds to chemical agents was discussed in depth in the following section when the toxicological potential of organic and inorganic compounds is taken into account. However, we agree that the reference to secondary ionic species can be misleading here, and will modify the text by just stating "Therefore, as discussed above, the higher ROS activity of daytime aerosols compared to that of nighttime/interstitial aerosols must be attributed to WS components, while WI compounds dominate ROS activity at night", and discuss the identification of potential redox-active chemicals in Section 3.2.

Page 7, Line 14-16: Although, it seems possible that the high correlation of inorganic species such as SO4-2, NO3-, etc. could be due to their co-linearity with the WSOC, however, recent studies (Environmental Science & Technology 51 (5), 2611-2620; Environ Sci Technol. 2012, 46(12):6637-44) have shown that SO4-2 enhanced acidity of the aerosol can solubilize the metals (such as Fe and Cu), which are known to generate ROS. Do the authors think to include such possibility in their work?

Reply: this is an interesting possibility. However, the acidity of fog is highly variable geographically, depending on the availability of atmospheric acidic (e.g., sulfuric, nitric and hydrochloric acids and acidic sulfates) and basis (ammonia) species. The historical trends of fog pH in the Po Valley (Giulianelli et al., 2014) show that the overall acidity has drastically decreased in the last 20 years, as a consequence of the reduction in SO$_2$ emissions, which was not accompanied by similar reductions in ammonia emissions that remained instead relatively constant with time. In recent years, fog pH in San Pietro Capofiume was around 6 or higher. Therefore, a possible effect on ROS from metal solubilization remains questionable in this specific environment. Nevertheless, we will add the following text to the paper to make clear to the reader that the attribution of ROS activity of fog to WSOC cannot be generalized, as multiple mechanisms can become important in different environments: "It should be noted, however, that inorganic acids can indirectly affect the ROS activity of fogs and wet aerosols by solubilizing redox-active metals (Oakes et al., 2012; Fang et al., 2017). Such mechanisms can be important for areas where SO$_2$ levels are high, while they are less plausible for environments such as the Po Valley where aerosols and fogs are largely neutralized by ammonia, as demonstrated by the typical fog pH of about~ 6 , according to Giulianelli et al., 2014)."

Page 8, Line 14: See my earlier comment, could the toxicity of Fe, Ni and Cu be due SO4-2 enhanced acidity and the solubility of these metals?

Reply: see the reply to previous comment.

Page 9, Line 4: There should be "with" between "water" and "respect"

Reply: will be corrected.

Page 10, Line 2: What do you mean by scavenging rate of ROS? I think this sentence needs to be either further cleared or modified.

Reply: we meant the scavenging rate of total redox-active compounds, as measured by ROS activity. We will rephrase the text.